# Crack Unet: Crack Recognition Algorithm Based on Three-Dimensional Ground Penetrating Radar Images

**DOI:** 10.3390/s22239366

**Published:** 2022-12-01

**Authors:** Jiaming Tang, Chunhua Chen, Zhiyong Huang, Xiaoning Zhang, Weixiong Li, Min Huang, Linghui Deng

**Affiliations:** 1Xiaoning Institute of Roadway Engineering, Guangzhou 510640, China; 2School of Civil Engineering and Transportation, South China University of Technology, Guangzhou 510641, China

**Keywords:** 3D ground penetrating radar, crack recognition, convolutional neural network

## Abstract

Three-dimensional (3D) ground-penetrating radar is an effective method for detecting internal crack damage in pavement structures. Inefficient manual interpretation of radar images and high personnel requirements have substantially restrained the generalization of 3D ground-penetrating radar. An improved Crack Unet model based on the Unet semantic segmentation model is proposed herein for 3D ground-penetrating radar crack image processing. The experiment showed that the MPA, MioU, and accuracy of the model were improved, and it displayed better capacity in the radar image crack segmentation task than current mainstream algorithms do, such as deepLabv3, PSPNet, and Unet. In the test dataset without cracks, Crack Unet is on the same level as deepLabv3 and PSPNet, which can meet engineering requirements and display a significant improvement compared with Unet. According to the ablation experiment, the MPA and MioU of Unet configured with PMDA, MC-FS, and RS modules were larger than those of Unet configured with one or two modules. The PMDA module adopted by the Crack Unet model showed a higher MPA and MioU than the SE module and the CBAM module did, respectively. The results show that the Crack Unet model has a better segmentation ability than the current mainstream algorithms do in the task of the crack segmentation of radar images, and the performance of crack segmentation is significantly improved compared with the Unet model. The Crack Unet model has excellent engineering application value in the task of the crack segmentation of radar images.

## 1. Introduction

The pavement structure of a road will inevitably receive damage, such as cracks, subsidence, and interlayer cavitation, under the effect of traffic load, temperature, ultraviolet light, etc., as its operation time lengthens. Therefore, it is very important for the engineers to determine the maintenance time and formulate a maintenance scheme scientifically and reasonably by detecting the location, type, and severity of the pavement structure damage quickly, effectively, and precisely and forecast the development of those damages [1,2,3,4,5,6,7,8,9,10,11]. Cracking is the most frequent among the various damages in the pavement structures. Therefore, it is urgent for the engineers to conduct fast, precise, and non-destructive testing technology research to obtain crack information inside the pavement’s structure.

Ground-penetrating radar detection is a major and representative technology for non-destructive road testing. Through this method, the engineers obtain information on an object by analyzing the propagation of electromagnetic waves inside the object. To detect the internal damage detection of old roads, the technology makes a visual diagnosis of the road’s health conditions and realizes the rapid quality inspection of the road’s structure, which is similar to a “physical examination through Computer Tomography (CT)”, and this subtracts the great effort of core drilling. Compared with traditional technologies, grounding-penetrating radar detection has the technical advantages of causing no destruction and having a high detection efficiency [12].

Among the ground-penetrating radars with various mechanisms, two-dimensional (2D) ground-penetrating radar displays low precision and accuracy in crack detection due to its single antenna setup. In contrast, 3D ground-penetrating radar collects multi-channel radar data simultaneously to form 3D radar data, and thus, it detects those cracks via multiple channels, jointly. It significantly reduces the misjudgment rate, and therefore, it is an effective approach for detecting cracks and damage inside the pavement’s structures.

However, in terms of 3D ground-penetrating radar crack image interpretation, there is currently a lack of automated recognition methods, and images are mainly interpreted manually. Many problems exist in manual radar image interpretation. For example, the interpretation process requires highly professional expertise, and there is a lack of data interpreters. The interpretation process is subjective; the same radar image often produces different results when it is interpreted by different people. The manual interpretation process includes the features of it requiring a long working period, and it having a huge workload, and a low efficiency level. Those disadvantages have hindered the application and promotion of ground-penetrating radar technology to a certain degree [13,14].

An improved Crack Unet model based on the Unet semantic segmentation model is proposed to tackle the problems in 3D ground-penetrating radar image crack recognition. Section 2 discusses the current progress in the relevant research and the necessity and significance of the research. Section 3 explains the data acquisition equipment and the principles of 3D ground-penetrating radar and the establishment of a model database. Section 4 details the structure, components, and design considerations of the Crack Unet model. In Section 5, the appropriate loss function and the image standardization method were selected based on the characteristics of radar image cracks, and a series of tests were conducted, such as the crack dataset test, the non-crack dataset test, the ablation test, and the attention module contrast test. Section 6 summarizes the key conclusions herein, and Section 7 proposes the tasks that are to be researched further.

## 2. Literature Review

The current methods for crack detection inside the pavement’s structures include core sampling, optical fiber sensing detection, ultrasonic detection, industrial CT scanning, and ground-penetrating radar. Among them, core sampling uses single-point detection, and it causes damage to the pavement, and the resultant data are poorly representative. Optical fiber sensing detection technology has the advantages of having a high precision and real-time monitoring, but it can only complete the detection over a small coverage area. Ultrasonic detection technology is mainly applied to the crack detection of cement concrete slabs, and it is greatly compromised in terms of crack detection inside asphalt pavement and sub-structural layers. Electromagnetic waves can be transmitted and reflected steadily in the pavement’s structures and materials because of their electromagnetic features. Ground-penetrating radar can obtain a degree of high precision and a large detection depth in terms of crack detection inside the pavement’s structures.

Lots of research has been conducted from the aspect of automatic radar image interpretation. Many methods have been proposed, such as canny edge detection, Hough transform, the support vector machine (SVM), compressed sensing, and convolutional neural networks. Among them, the cracks can be segmented accurately through canny edge detection, but there are many false alarm signals, and the recognition accuracy is low because edge detection only takes into account the local gradient change in the images [15]. Hough transform is an effective method for engineers to detect and locate straight lines and analytical curves, and its algorithm accuracy is less vulnerable to random noise and signal discontinuity. However, the Hough transform is inapplicable to the automatic recognition of crack signals because those cracks do not have the morphological characteristics of fixed signals [16]. SVM requires a large number of training samples for image feature selection and model parameter optimization, which are mainly used for region categorization, but it cannot segment the crack edge accurately.

The convolutional neural network, which has emerged in recent years, can complete the task of recognizing those similar features in other unmarked images by learning from the correctly marked images [17,18,19,20,21]. Haifeng Li, et al. proposed a two-stage recognition method for GPR-RCNN [22]. This method has achieved the highly precise recognition of targets such as cavitation, pipelines, and subsidence. However, its accuracy and misjudgment rate for crack recognition are significantly lower than those of the remaining three targets. Liu, W.C., et al. designed a CNN model with a simple structure, which was called the visual GPR (VGPR), and it was set to train, validate, and test the VGPR with the GPR image data. In the test, the weighted average F1 score in the comprehensive metrics reached 99.626%, indicating that VGPR has an excellent generalization performance [23]. Yamaguchi, T proposed a unique method for Electro Magnetic simulation and a deep learning technique to consider the 3D reflection patterns of voids. The hyperboloid reflection patterns of voids were extracted by the 3D convolutional neural network (3D-CNN). The accuracy of 3D-CNN classification reached 90%, which was by about 10% compared to the previous 2D-CNN, which demonstrates its effectiveness in 3D subsurface sensing and detection. At present, the research of GPR image recognition mainly focuses on the intelligent recognition algorithm based on object detection [24]. The recognition occurs in a rectangular area, and the method fails to achieve the accurate segmentation of the cracks.

The semantic segmentation method can achieve the purpose of separating the cracks accurately. The common semantic segmentation algorithms include DeepLab [17], Unet [25], PSPNet [26], and so on. Many researchers have applied the semantic segmentation model to the interpretation of the GPR data. Dai, Q.Q. et al. proposed a two-stage deep neural network (DNN), called DMRF-UNet, and they reconstructed the permittivity distributions of the subsurface objects from GPR B-scans under heterogeneous soil conditions [27]. In the first stage, the U-shape DNN with the first multi-receptive field convolution group (MRF-UNet1) was built to remove the clutters due to the inhomogeneity of the heterogeneous soil. Then, the denoised B-scan from MRF-UNet1 was combined with the noisy B-scan to be inputted to the DNN in the second multi-receptive field convolution group (MRF-UNet2). Luo, S.G. et al. proposed an A-Unet deep learning network that was designed to achieve underground target imaging [28]. The input data have a multi-scan MF amplitude, and the phase data were extracted from the GPR B-Scan data, while the output was the underground dielectric parameters distribution in a designated regime. Numerical simulation experiments have proved that this method effectively reconstructs the shape distribution of the underground targets, and the training time of the added structure was shortened to 9.09% of the training time in the skip-connection unit without reducing the imaging resolution.

Although many researchers have applied the semantic segmentation algorithm to GPR image interpretation, it has not been applied to GPR image crack signal extraction. The semantic segmentation adopts the complete convolutional neural network which categorizes each pixel in an image to segment the crack accurately. This is the most necessary function for the engineers to interpret the 3D radar images and extract the crack features.

An optimized semantic segmentation model, namely, the Crack Unet based on Unet model is proposed herein, which was applied to 3D GPR crack signal segmentation. The Crack Unet model is based on the Unet model, and it incorporates the PMDA, MC-FS, RS module and PMDA module. The model has shown a better capacity than the mainstream models have such as deepLabv3, PSPNet, and Unet in the radar image crack segmentation.

## 3. Data Collection and Database Establishment

The 3D ground-penetrating radar system was adopted in this research. It mainly consists of the host, the real-time dynamic positioning system, and the multi-channel antenna array, as shown in Figure 1a.

One can refer to the reference [29] for the 3D ground-penetrating radar data acquisition and preprocessing stages. During the detection, the trigger distance parameter was set at 0.03 m, the depth range was set at 62.5 ns, and the dwell time was set at 1ms, and the antenna combination configuration is shown in Figure 1c. Each scan of the multi-channel antenna array was as wide as 1.5 m, as shown in Figure 1b. With the multiple set detection channels, multiple strings of data can be spliced in combination with accurate subsequent positioning data to achieve the complete section coverage scanning of any road.

The data collected by transmitting and receiving antennas at a certain measuring point represent a one-dimensional data set, namely, it is called A-Scan, as shown in Figure 2a.

When the antenna moves forward, the distance of the device is recorded through the distance encoder (DMI), and the antenna is triggered equally to collect data, forming two dimensions in the driving direction and the depth direction, namely, the B-Scan, as shown in Figure 2b.

The 3D GPR antenna has 20 pairs of antennas, which can collect 20 B-Scans at the same time. The combination of multiple horizontal slices can form 3D data with the driving direction, the transverse direction, and the depth direction, namely, the C-Scan, as shown in Figure 2c.

In the process of the C-scan data analysis, the sections can be cut from three directions: as shown in Figure 3, in the horizontal section (XY Plane, green), the longitudinal section (XZ Plane, purple), and the transverse section (YZ Plane, red) to analyze the data. During the process of the radar data recognition, the 3D radar data (C-Scan) were transformed into 2D image data (B-Scan) in a form of slices to ease recognition. The 3D coordinate system was established with the traveling direction of the radar antenna in the *X*-axis, the depth direction perpendicular to the ground in the *Z*-axis, and the direction perpendicular to the X–Z plane in the *Y*-axis.

Before the recognition of the 3D ground-penetrating radar images, the data of the 3D ground-penetrating radar need to be preprocessed by 3D radar examination. The processing steps include the interference suppression, ISDFT (Inverse Fourier Transform), BGR (high pass), and Auto scale ones. The parameter settings are shown in Figure 4.

The horizontal spacing (Y) of the data matrix collected by 3D radar was 0.071 m, the spacing in the traveling direction (X) was 0.025 m, and the spacing in the depth direction (Z) was about 0.009 m. In order to homogenize the data matrix, it was necessary to interpolate the horizontal (Y) and traveling directions (X) so that the spacing of the two directions was close to the depth direction and that the spacing among the points in the three dimensions of X, Y and X was 0.009 m. The interpolation method adopted a bilinear interpolation, and the interpolation calculation method is shown in Figure 5.

For the given adjacent detection data points P1, P2, P3, and P4, the point was inserted into any position of the rectangular area with the four points as the corner ones, and the function value at this point was calculated as follows:(1)$$f=\frac{\left(x_{2}-x)(y_{2}-y\right)}{\left(x_{2}-x_{1})(y_{2}-y_{1}\right)}f_{1}+\frac{\left(x_{2}-x)(y-y_{1}\right)}{\left(x_{2}-x_{1})(y_{2}-y_{1}\right)}f_{2}+\frac{\left(x-x_{1})(y-y_{1}\right)}{\left(x_{2}-x_{1})(y_{2}-y_{1}\right)}f_{3}+\frac{\left(x-x_{1})(y_{2}-y\right)}{\left(x_{2}-x_{1})(y_{2}-y_{1}\right)}f_{4},$$
In the Equation (1), $x_{i}$ and $y_{i}$ are the coordinates of Pi (i = 1, 2, 3, 4), $f_{i}$ is the radar data value of Pi (i = 1, 2, 3, 4), as shown in Figure 5, x and y are the coordinates of the points whose radar data values needed to be calculated, and f is the calculating radar data value.

From Equation (1), the value of the radar data at any coordinate can be calculated. After the data were encrypted and interpolated, the crack signal was performed in the three perpendicular slices, and this is shown in Figure 6a. It can be seen that the crack signal is not obvious on the Y–Z plane and X–Z plane, but it is relatively obvious on the X–Y plane. That is relevant to the longer length and the generally narrow width and low height of the cracks. Therefore, the X–Y plane of the 3D radar data was intercepted for the crack recognition.

The dataset adopted herein consists of the X–Y plane of the ground-penetrating radar. After each test lane was scanned, their images needed to be spliced for a complete radar image of the road. The cut result is shown in Figure 6b. Since some of the areas may be missed during separate lane scanning, the image may be left with black areas. Each radar signal area divided by those black areas corresponds to the lane. In addition, the vehicle may encounter obstacles during as it drives so the radar image of each lane may be curved or contain overlapped areas. However, most of the effective radar information will not be compromised, and the crack segmentation can still be performed.

A large number of radar images on the horizontal road plane were obtained using the above method. An appropriate number of crack images and undamaged images were selected from those images to form the dataset. The crack signal in the radar image is the same as what is shown on the road chart. The color is dark or bright, and the area is significantly differentiated from the surrounding area. During the selection, it should be ensured that the dataset contains various cracks with the common shapes, including transverse cracks, longitudinal cracks, and cross cracks. Figure 7 shows the radar images under various working conditions in the dataset. Since the traveling direction of the radar vehicle was longitudinal, the longitudinal crack was displayed as the horizontal crack, and the transverse crack was displayed as the longitudinal crack in the Figure 7.

The dataset adopted herein contains 4915 images, and each of them has a resolution of 604 × 604 which is represented the actual range, 5.436 m × 5.436 m. After the selection, the Labelme tool was employed to label the crack area in each image as shown in Figure 7d. Since the radar image is different from the common RGB image, there may be more interference signals, or the edge pixels of cracks may not be accurately determined due to the impact of different working conditions. Furthermore, the black area in the middle was separated into two parts since this was the vacant area during the splicing, and it did not belong to any part of the cracks. Finally, 3915 images in total were adopted as the training set and the validation set, with the ratio of 8:2. Among them, the validation set was not involved in the training, but it was used to verify the results from each round of training. The remaining images that served as the test set were divided into two parts. The first part was the crack test set. There were 500 images in total, and each contained cracks. They were adopted to test the model’s ability to segment cracks. The remaining 500 images were crack-free test set. They were adopted to test whether the model was prone to make misjudgments.

## 4. Crack Unet Algorithm

The Crack Unet employed herein is an improved Unet semantic segmentation model based on VGG16. The structure of Crack Unet is shown in Figure 8. The gray rectangle in Figure 8 is the characteristic pattern output at each step. The numbers marked near the gray matrix are the width, height, and latitude of the matrix, respectively, representing the 3D matrix of 512 × 512 × 3. The arrows in different colors represent the processing steps for different characteristics. These processing steps are explained in Table 1, and the important optimization steps are explained in detail in Section 4.1, Section 4.2 and Section 4.3.

The network was divided into the left part and the right part, with the left half being the encoder and the right half being the decoder. The encoder was divided into five modules, each of which contained a multi-channel convolution feature selection (MC-FS) module, a parallel multi-direction attention mechanism (PMDA) module, and a residual compression (RS) module. The size for the image input in the model was 512 × 512, which was chosen to facilitate the down-sampling in multiples of 2 and to make the size as close as possible to that of the original image so to avoid feature loss due to the image reduction process.

The MC-FS module in the network was placed in the first position of each module to increase the number of channels and simultaneously extract multiple features. Then, those features were selected to ensure the validity of the features input in the subsequent operation, and thus, to improve the coding capacity of the encoder. The 3 × 3 convolution kernel was selected for further feature extraction in the model. This is because the MC-FS module contains the expanded convolution, and it already has a large receptive field. In this process, the small convolution kernel can reduce the overall parameter number of the model. Then, the PDMA mechanism module will establish the correlation among the feature pixels so to weigh the crack pixels and strengthen the capacity for the network to extract the crack features. The final output features perform maximum pooling to reduce the feature resolution.

The feature coding was completed as the above steps were performed five times in a row. The encoded data entered the decoder from the last layer of the encoder. The output features of the last layer of encoder were upsampled. The resolution became twice as large as the original size was, and it formed the feature of the decoder. This feature was then connected to the output feature of the corresponding encoder. At this moment, the feature of the encoder would first enter the compression module for channel compression to remove unnecessary shallow features and simultaneously reduce the network scale before its connection. In this way, information loss can be prevented. Then, two feature extractions were performed. The decoding was completed as the above steps were performed four times. The resolution of the decoded characteristic pattern was the same as that of the original image, with the channel number being the category number. Then, softmax activation was performed on all of the channels of each pixel to complete the pixel-level categorization. The PDMA mechanism, the MC-FS module, and the RS module are introduced in Section 4.1, Section 4.2 and Section 4.3, respectively.

### 4.1. Parallel Multi-Direction Attention Mechanism

The attention mechanism is a commonly employed lightweight module for network performance improvement, whose principle is to weigh specific dimensions of the features. The weights are self-learned by the neural network, and therefore, the process requires no human intervention [26]. The parallel multi-direction attention (PMDA) mechanism proposed herein is a spatial attention mechanism, whose principle is to break down a 3 × 3 convolution kernel into four parts, and then, correlate a pixel with its adjacent pixels artificially from the structure. The structure is shown in Figure 9.

The PMDA can be divided into four parts. From the top to the bottom, in terms of the input Feature A with the shape of H × W × C, the first two parts adopt a 2 × 1 kernel and a 1 × 2 convolution kernel, respectively, to establish the correlation between each pixel in Feature A and its left and lower pixels; the two parts of this convolution obtain Features B and C, respectively. The third and the fourth parts adopt a 2 × 2 convolution kernel with an expansion rate of 2 and a 1 × 1 convolution kernel. Those two parts work jointly to establish the correlation between each pixel and its four adjacent pixels on its two diagonals to obtain the integrated Feature D1 and D2. Therefore, the “Add” operation is performed on the results of those two parts to obtain the integrated Feature D. Then, the 2 × 2 convolution kernel is adopted to connect the results of Features B, C, and D to combine the channels of Feature F to obtain Feature G; the sigmoid activation function is then employed to convert its value into the weight of the (0, 1) interval. Finally, Feature G is multiplied with the original characteristic of Feature A to complete the pixel weighting in the characteristic pattern and to obtain Feature A’. Since only five convolution kernels are employed and they are very small, the attention module only adds a few parameters, and thus, it is easy to realize.

The PMDA module is designed on the basis of crack shape characteristics because the cracks are generally thin and continuous. If a pixel is a crack pixel, it is highly possible that there are crack pixels among its adjacent pixels; if the pixel is the background, then the adjacent area is most likely the background as well. Therefore, the correlation among the pixels is more conducive to crack feature extraction. Moreover, this module is beneficial to the difference between the crack and the structural crack. Since the cracks are irregular, while structural cracks are straight, the feature pixels of the structural cracks usually lack the correlation in the diagonal position. PMDA is beneficial to this feature extraction when it correlates the pixels. The PMDA module has been added to each module of the encoder in the Crack Unet model that is adopted herein to improve the capacity of the crack feature extraction of the encoder.

### 4.2. Multi-Channel Convolution Feature Selection Module

The multi-channel convolution feature selection module (MC-FS Module) can provide diverse features and give higher weight to the effective features in the dimension of the channel. The MC-FS module consists of the parallel convolution, extended convolution, and channel attention modules. Its structure is shown in Figure 10, in which w, h and c represent the width, the height and the number of channels features input, respectively, and C represents the number of output channels.

Among the four convolutions first performed in parallel for the input features, two of them are 3 × 3 and 2 × 2 expanded convolutions with an expansion rate of two. They aim to obtain a larger receptive field with a small number of parameters. The addition of a standard 3 × 3 convolution and a 2 × 2 convolution has prevented the checkerboard effect of expanded convolution, and it has eliminated the problems such as the discontinuity and the local feature loss during the feature extraction by expanded convolution. Therefore, for one input feature, the MC-FS module generates four features. The channel number for each feature is 1/4 of the output feature. The output channel number were restored after those four features were connected.

Finally, the channel attention module SENet (Sequeeze-Excitation) was added to weigh each channel to realize the feature selection and to give higher weight values to the effective channels. We performed a compression operation. $F_{\mathrm{sq}}$ on the feature map $U\in R^{H\;\times \;W\;\times \;C}$. For global pooling, the feature shape $F_{\mathrm{ex}}$ was compressed to 1 × 1 × C, and then, the activation operation $F_{\mathrm{ex}}$ was performed. Usually, the utilization of the ordinary convolution or complete connection layer is performed to learn the weight of the output channel. Finally, the characteristic pattern U was weighted to produce the characteristic pattern $\tilde{X}\in R^{H\;\times \;W\;\times \;C}$. Similarly, channel attention does not change the feature shape before and after its use.

There are three benefits to incorporate the MC-FS module: (1) According to Figure 10, the MC-FS module is used in the first part of each sub-module of Crack Unet to increase the feature channel number. (2) The combination of expanded convolution and standard convolution captures both the large and small receptive fields simultaneously, which avoids the checkerboard effect and reduces the parameter number. (3) Four-way parallel convolution makes the extracted feature combination more diversified. The feature weighting of the channel attention module diversity can be used to improve the accuracy and effect of the extracted features.

### 4.3. Residual Compression Module

The residual compression module (RS) was designed to compress the feature channel and reduce the number of network model parameter to a quarter of that which it was before. The module was used for those shallow features of the encoder before the residual connection step [30]. In Crack Unet, the decoder will connect the features of the encoder on the corresponding layer as a supplement. However, since the encoder belongs to the shallow region and it contains a large amount of “rough” information in comparison with the decoder, the connection among all of these features requires the decoder to “refine” these features when it supplements the information. In fact, not all of the information is valid. In this way, such a connection does nothing more than to add a great amount of noise to the deep features. Therefore, only the appropriate information needs to be supplemented to the decoder, and there is no need to connect all of the features to the decoder. Therefore, the RS module is designed to remove the redundant channels to reduce parameter number. The RS module is very simple in its structure, as shown in Figure 11.

After entering the compression module, the features were maximally pooled at a step of two, and the feature values with the largest response were retained. Then, the resolution would be restored through de-pooling. Such an operation was performed to reserve the most spatially efficient features for encoder connection. Finally, the 2 × 2 ordinary convolution was adopted to change the feature channel. The compression rate herein was 0.25. That is to say, the channel number was compressed to 1/4 of the original number.

## 5. Experiment and Result Analysis

### 5.1. Experimental Configuration

For the training configuration employed herein, the GPU model was NVIDIA 1080Ti; the video memory size was 11 GB; the operating system was the computing platform of Center OS7; the deep learning framework adopted the Keras with TensorFlow-GPU as its back-end. During the training process, random level flipping, translation, and scaling were employed to enhance the dataset.

Before the model started training, there were still some parameters that needed to be set. The learning rate is a tuning parameter in the optimization algorithm that determines the step size at each iteration while moving toward the minimum of a loss function; the initial learning rate was set to 10^−5^.

One training round refers to the model completing all of the image recognition and model modification processes. The max training round of the model was set to 200. The mechanism of automatic reduction and early termination of the learning rate was adopted. That is to say, the validation set loss was monitored. If the validated loss did not decrease within six rounds, the learning rate would be halved. If it lasted for 10 rounds, the training would be terminated.

### 5.2. Loss Function

Due to the slender feature of the cracks, its pixels account for a small proportion of the image. Therefore, Dice Loss and Focal Loss were employed as loss functions. Dice Loss was Proposed in VNet, which was initially adopted to solve the problem of sample imbalance in medical image segmentation. The formula was defined as follows:(2)$$Loss_{dice}=\frac{2\sum_{i=1}^{N}p_{i}y_{i}}{\sum_{i=1}^{N}p_{i}^{2}+\sum_{i=1}^{N}y_{i}^{2}},$$

In Equation (2), $y_{i}$ is the label of the ith pixel in the image, $p_{i}$ is the prediction result of the ith pixel, and *N* is the pixel number in the image.

Focal Loss was also employed to solve the problems of unbalanced training samples of different difficulties in the sample, which was a variant of cross entropy. The Focal Loss of a pixel in semantic segmentation was defined as follows:(3)$$Loss_{focal}=-{\left(1-p_{t}\right)}^{\gamma }\log(p_{t}),$$
In the formula, *γ* is a constant of 2 and $p_{t}$ is the prediction probability of the model for samples with positive prediction results, which is defined as follows:(4)$$p_{t}=\left\{ \begin{array}{l} p,y=1\\ 1-p,otherwise \end{array}\right.,$$
Then, the loss function that was employed in the model was $Loss_{dice}+Loss_{focal}$.

### 5.3. Image Standardization

The images were standardized, and then, centralized by a mean value to improve the generalization ability of the trained model and the training efficiency. The data after the centralization are more in line with the distribution law; their calculation is shown in Equation (5) as follows:(5)$$X^{\prime }=\frac{X-\mu }{\sigma },$$
where *X* is the image matrix, μ is the mean of the matrix, and σ is the standard deviation of the matrix.

The decline curves of the training set loss and validation set loss are shown in Figure 12a,b. Since the automatic stopping policy was set, the model converged at the 63rd epoch.

### 5.4. Experiment Evaluation Indexes

The index category from the semantic segmentation standard was adopted as the evaluation index of the model, including the Mean Pixel Accuracy (MPA), the Mean Intersection over Union (MIoU), Floating-point Operations (FLOPs), and the FPS. Among them, MPA is the proportion of those correctly categorized pixels to all of the pixels, which can be taken to evaluate the accuracy of pixel-level categorization. MPA is equal to the mean value of the sum of PA in all of the categories. For a category in semantic segmentation, the pixel accuracy (PA) is:(6)$$\mathrm{PA}=\frac{\mathrm{TP}+\mathrm{TN}}{\mathrm{TP}+\mathrm{FP}+\mathrm{FN}+\mathrm{TN}},$$
In Equation (6), TP, TN, FP, and FN are the classification of the accuracy of pixel segmentation results. Each pixel in the segmentation result corresponds to one of the following four categories:

(1) True Positive (TP): The model prediction is true positive, and the label is true positive;

(2) False Positive (FP): The model prediction is true positive, while the label is false positive;

(3) False Negative (FN): The model prediction is false negative, while the label is true positive;

(4) True Negative (TN): The model prediction is negative, and the label is also negative.

In addition to the target category that needs to be segmented, semantic segmentation also has the default category of the background. Therefore, there are two types of segmentations herein: the crack and the background. IoU is the intersection of the labeled region and the predicted region divided by their union set. That is to say, for the label region A and the prediction region B, the IoU is expressed as:(7)$$IoU=\frac{A\cap B}{A\cup B}=\frac{\mathrm{TP}}{\mathrm{TP}+\mathrm{FP}+\mathrm{FN}},$$
In Equation (7), the definitions of TP, TN, FP, and FN are the same as they are in Equation (6). Then, MIoU is the mean value of the sum of the IoU values for all of the categories. IoU can evaluate how well the segmentation region overlaps the label expectation. FLOPs represent the number of floating point operations of the neural network, and they can be adopted to evaluate the computation workload or complexity of the model. The FLOPs for a single convolutional layer were calculated as follows:(8)$$\mathrm{FLOPs}=(2\times C_{\mathrm{in}}\times K^{2}-1)\times H\times W\times C_{\mathrm{out}},$$
where $C_{\mathrm{in}}$ is the number of input channels, K is the size of convolution kernel, H and W are the height and width of the output characteristic pattern, respectively, and $C_{\mathrm{out}}$ is the number of output channels.

The accuracy was incorporated as another index in order to more intuitively represent the segmentation capacity of the model and the accuracy of the reaction model at the image level. The radar image of crack would be interfered with by electromagnetic waves from various sources and the boundary ambiguity of the crack in the road, as shown in Figure 13:

The cracks in the image extend to a blurry position, and then, they disappear so it is impossible for the model to accurately label the crack. Only the areas with significant signals can be marked out as carefully as possible during labeling. However, in the fuzzy area, the crack area should be covered as much as possible, and the crack edge should be ignored accordingly. Due to the uncertainty of image labeling, such images will introduce difficulties to the model training so that false positive pixels relative to the labeling will be generated during the test. However, those pixels that are mis-classified by the network as cracks are reasonably relative to the original image. Therefore, it was not appropriate to use PA or IoU as indexes at this moment. Therefore, the accuracy of the model in the test set can be evaluated according to the coverage of the segmentation results to the labeled area, and the coverage rate is much more intuitive and simpler than the IoU is. During the test of crack test set, it is stipulated that an image should be considered to be successfully segmented as long as the coverage rate of its segmentation area relative to the labeled area is larger than 50%. Therefore, the formula of coverage rate was expressed as:(9)$$\mathrm{Accuracy}=\frac{A\cap B}{A}=\frac{\mathrm{TP}}{\mathrm{TP}+\mathrm{FN}},$$
where A is the label area and B is the segmentation area of the network. The coverage rate can reflect the integrity of the model segmentation. However, in the test of the non-crack data set, the model accuracy could not be reflected by the coverage rate since there was no crack in the test set. Therefore, the proportion of pixels classified as cracks to all of the pixels in a single image was used to reflect the accuracy of the non-crack data set. When the proportion between the two exceeds a certain threshold, the categorization will be considered to be wrong.

### 5.5. Experiment and Result Analysis

The currently mainstream models and modules were taken to make a comparison with Crack Unet and its modules. The experiments include the model comparison between the Crack Unet and the mainstream semantic segmentation networks, the ablation experiment between the Crack Unet and the PMDA, MC-FS and RS modules, and the comparison experiment among the PMDA, SE, and CBAM modules.

#### 5.5.1. Model Comparison

The currently mainstream algorithm models of the semantic segmentation were taken to compare them with Crack UNet using the semantic segmentation indexes in the test set. The test was conducted using the crack data set. The models involved included deepLabv3, PSPNet, and Unet. All of the test environments were the same as they were in the training environment. The test results are shown in Table 2.

According to the table, both the MPA and MIoU of the method adopted that was herein are higher than those in other models. Among them, the Crack Unet MPA was 7.35%, 2.27%, and 3.6% higher than that of DeeplabV3, PSPNet, and Unet, respectively. The Crack Unet MIoU was 1.4%, 0.99%, and 2.27% that of DeeplabV3, PSPNet, and Unet, respectively. They were all substantially improved, and the FLOPs value of the model was also the lowest. PSPNet had the fastest detection speed, and the MPA and MIoU values were closest to those of the Crack Unet ones. Although the FPS of Crack Unet was still low and the segmentation speed was slow, it also reached the requirement for real-time detection. More attention has been paid to the damage recognition capacity during the practical engineering application of road detection. Therefore, the method herein has a better segmentation capacity in the radar image crack segmentation task than the currently mainstream algorithms do. The pixel-level categorization indexes are shown in the Table 3. 

The statistics on the segmentation results of each model on the test set and the coverage rate distribution of the labeled areas was also made to more intuitively display the image-level recognition performance of each model. When the coverage rate reached 50% or above, the current image segmentation can be considered to be successful. Therefore, the coverage rate reflects the crack segmentation integrity of the model. The distribution results of each model under different coverage rates are shown in Table 2.

According to the table, the accuracy of those four algorithms was 78.6%, 87.0%, 84.4%, and 93.0%, respectively. Deeplabv3 had the lowest accuracy and most of those segmentation results had a coverage rate of (0.6, 0.7). The other algorithms produced higher coverage rates. The coverage rate of Unet was mostly concentrated in the interval of (0.6, 0.9), while the coverage rates of PSPNet and our method were concentrated in the interval of (0.7, 1.0). Therefore, in terms of the segmentation integrity, Crack UNet was the optimal one, and it had the highest accuracy of 93.0%. The segmentation results of each model are shown in Figure 14. According to Figure 14, DeepLabv3 and UNet had segmentation errors or incomplete segmentation; their segmentation results were relatively slender and long, which means that they were prone to the erroneous segmentation of crack pixels. PSPNet and our method delivered relatively complete segmentations and a large width so their coverage rates were higher, and their integrity degrees were better. Due to the comparison, it can be observed that Crack UNet produced better results.

In order to test the model accuracy under the non-crack condition, all of those models were tested using the non-crack data set. In addition, the accuracy of each model was calculated at thresholds of different false alarm percentages using the non-crack data set. The false positive percentage herein was the ratio of that of the mis-categorized pixels in an image to the total pixels. Three thresholds were adopted in the experiment, namely <1%, <0.5%, and <0.1%. The test results are shown in Table 4, below.

Unet produced the lowest accuracy at each threshold during the experiment with the non-crack data set. Although the accuracy rate of 88.80% that was achieved by Crack UNet was not the highest, it was not far from the accuracy rates that were achieved by DeepLabsv3 and PSPNet. In light of the precision of two datasets, the performances of DeepLabv3 and Unet were not stable. DeepLabv3 performed poorly in the crack test set but best in the non-crack test set. Therefore, DeepLabv3 had a poor segmentation capacity. However, the opposite was true for Unet. It performed well in the crack test set, but it produced more false positives in the non-crack test set. Therefore, Unet had a strong capacity in the segmentation test, but it was prone to erroneous categorization and had a low level of robustness. PSPNet was not only stable, but it was also accurate. However, PSPNet showed a poor accuracy in the crack test set. Although Crack UNet did not perform best in the non-crack test set, the gap between the DeepLabv3 and PSPNet results was small.

The segmentation results by each model using the non-crack test set are shown in Figure 15. Similarly, there were many false positives in Unet, and the areas with false positives were sparse and scattered. Although DeepLabv3, PSPNet, and Crack UNet produced false positives, they were relatively stable, and their false positives were few in number. Among them, the false positive areas of DeepLabv3 and PSPNet were continuous, while those of Crack UNet were sporadic and scattered, and their coverage was relatively smaller.

#### 5.5.2. Ablation Experiment

The network employed herein was obtained from the Unet modified by VGG16, so Unet was taken as the baseline for ablation experiment. The test indexes of various models in the crack data set were compared after the baseline was equipped with different modules. The experimental results are shown in Table 5. According to the table, MPA increased by 0.9%, 3.36%, and 1.7%, and MIoU increased by 0.93%, 0.8%, and 1.63%, respectively, after the PMDA, MC-FS, and RS modules were separately added to the network with Unet as the baseline. It showed that each module independently improved the performance of Unet. When Unet was equipped with two of three modules, MPA and MIoU also increased in comparison with the baseline, and MPA significantly increased. When the baseline was equipped with three modules, and thus, it became the model proposed, MPA and MIoU were the largest. According to the comparison of FLOPs and FPS values among each model, the models equipped with an RS module had lower FLOPs values than those without the modules did. Therefore, the FLOPs value of Crack Unet was at a low level. However, the FPS of Crack Unet was also lower, and it can be observed that the FPS values of the models equipped with the MC-FS module were similar. This indicates that MC-FS was a time-consuming module, while other modules had almost no impact on the models’ efficiency.

#### 5.5.3. Attention Model Comparison

The PDMA, SE, and CBAM modules were adopted to conduct a test using the crack test set, and their results were compared in order to verify that the model with the PDMA module had a better road crack segmentation capacity. That is to say, the MPA, MIoU, FLOPs, and FPS values were also employed as performance indexes, and these were used for the test of the crack test set. The comparison results are shown in Table 6.

According to the above table, the PMDA module produces higher MPA and MIoU and lower FLOPs values than either the SE module or the CBAM module did in both of the networks. Although the speed of PMDA is slightly slower than that of SE module, the difference is small. Holistically speaking, the PMDA module takes into account both the speed and accuracy, and so it is better in this task.

## 6. Conclusions

The Crack Unet model, which is an improved Unet semantic segmentation model based on VGG16, was proposed herein for 3D ground-penetrating radar image crack processing. The experimental conclusions of the model are as follows:

(1) The same crack data set was adopted for training and validation. Compared with three types of mainstream models, namely, the deepLabv3, PSPNet, and Unet ones, the experimental results herein showed that Crack Unet saw greater improvements than the other models did in terms of MPA, MIoU, and accuracy. Among them, MPA was 7.35%, 2.27%, and 3.6% higher, MIoU was 1.4%, 0.99%, and 2.27% higher, and the accuracy was 14.4%, 6.0%, and 8.6% higher than those of Deeplabv3, PSPNet, and Unet, respectively. Crack Unet also had the lowest FLOPs values. The FPS of Crack Unet was low, and its segmentation speed was slow, but the model reached the requirement for real-time detection. In the case of real-time detection, more attention has been paid to the damage recognition capacity during the practical engineering application of road detection. Therefore, the method herein has better segmentation capacity in the radar image crack segmentation task than the currently mainstream algorithms do.

(2) In the experiment using the non-crack data set, with the threshold of the proportion of false positive pixels being smaller than 0.1%, the accuracy rates of Crack Unet, deepLabv3, PSPNet, and Unet were 88.80%, 92.80%, 90.80%, and 53.80%, respectively. The accuracy of Crack Unet was slightly lower than those of the deepLabv3 and PSPNet models. However, they were on the same level. Crack Unet meets engineering requirements, and we saw a significant improvement over the Unet model.

(3) With Unet as the baseline of the ablation experiment, the results showed that in comparison with the baseline that was equipped with one or two modules, MPA and MIoU were the largest after the baseline was equipped with the PMDA, MC-FS, and RS modules. The model equipped with the RS module had a lower FLOPs value than it did without the RS module. MC-FS was the main time-consuming module, while other modules had almost no impact on the models’ efficiency.

(4) In the comparison test among the mainstream attention modules, the PMDA module produced higher MPA and MIoU and lower FLOPs values than either the SE module or the CBAM module did in both of the networks. Although the speed of PMDA was slightly slower than that of SE module, the difference was small, and both modules can meet the requirements for real-time data processing.

(5) The tests results show that the Crack Unet model has a better segmentation ability than the current mainstream algorithms do in the task of crack segmentation of radar images, and the performance of crack segmentation was significantly improved in comparison with the Unet model. The accuracy of the Crack Unet model reached 93.0%, which basically meets the engineering requirements of the crack segmentation of radar images. The Crack Unet model has excellent engineering application value in the task of crack segmentation of radar images.

## 7. Contents for Further Research

The experimental results from the crack data set show that the edge features of the structural cracks in those images, such as the bridge expansion joints and the joints between the slab and pavement base, are the same as those of cracks, and their signals are pretty strong. As a result, the semantic segmentation model will inevitably produce false positives in terms of these features. The radar images with structural cracks are different from those images under normal working conditions. To solve this problem, the engineers can employ classifiers to distinguish the two kinds of radar images to reduce the false positive rate of the model.

## Figures and Tables

**Figure 1 sensors-22-09366-f001:**
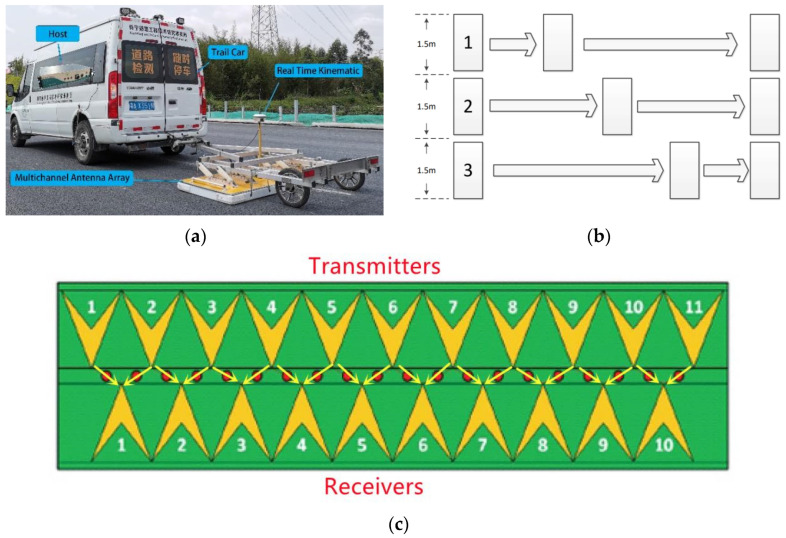
Onboard three-dimensional ground penetrating radar system and detection principle. (**a**) Onboard three-dimensional ground penetrating radar system. (**b**) Schematic diagram of 3D ground penetrating radar site detection (1.5 m means the width of detect line, numbers (1–3) mean the serial number of detect line). (**c**) Antenna combination configuration diagram (Numbers (1–11) mean the serial number of transmitters or receivers and yellow arrows mean the direction of electromagnetic wave propagation).

**Figure 2 sensors-22-09366-f002:**
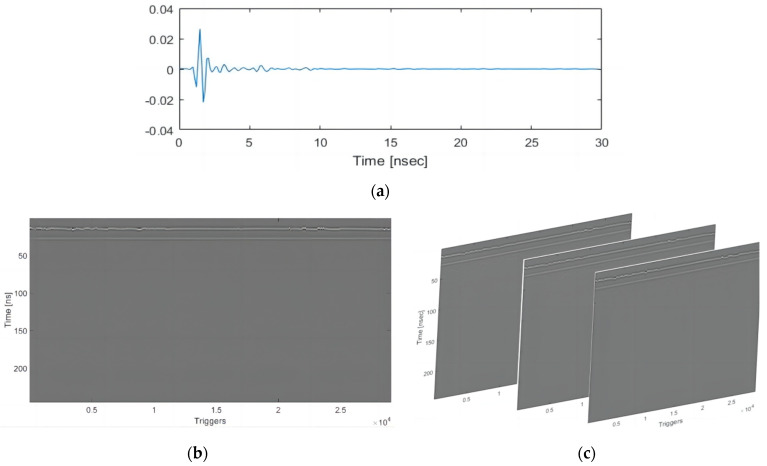
Ground–penetrating radar data in different dimensions. (**a**) A-Scan. (**b**) B-Scan. (**c**) C-Scan.

**Figure 3 sensors-22-09366-f003:**
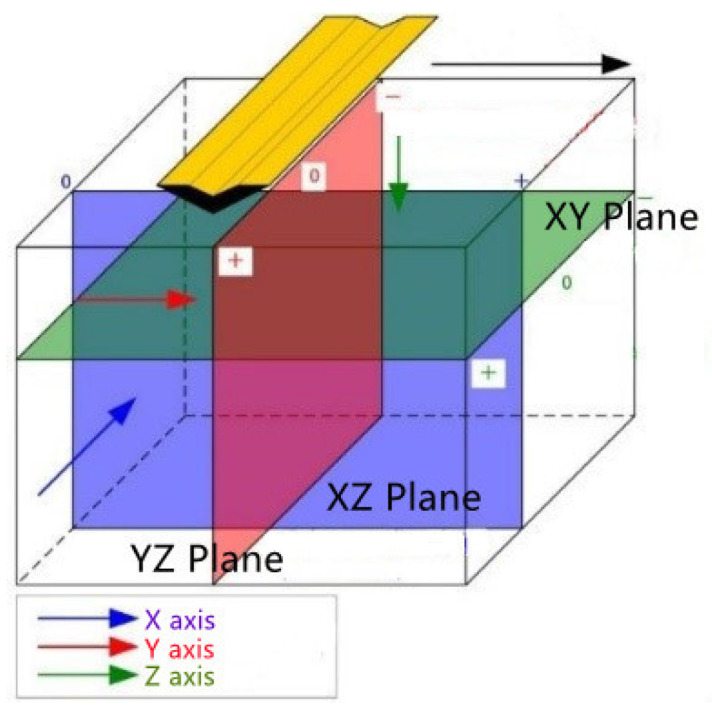
Three slicing methods for 3D ground-penetrating radar data.

**Figure 4 sensors-22-09366-f004:**
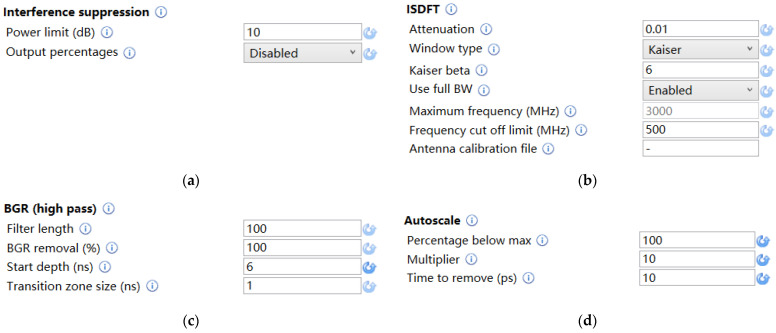
The parameter settings of preprocessing. (**a**) Interference suppression. (**b**) ISDFT (Inverse Fourier Transform). (**c**) BGR (high pass). (**d**) Autoscale.

**Figure 5 sensors-22-09366-f005:**
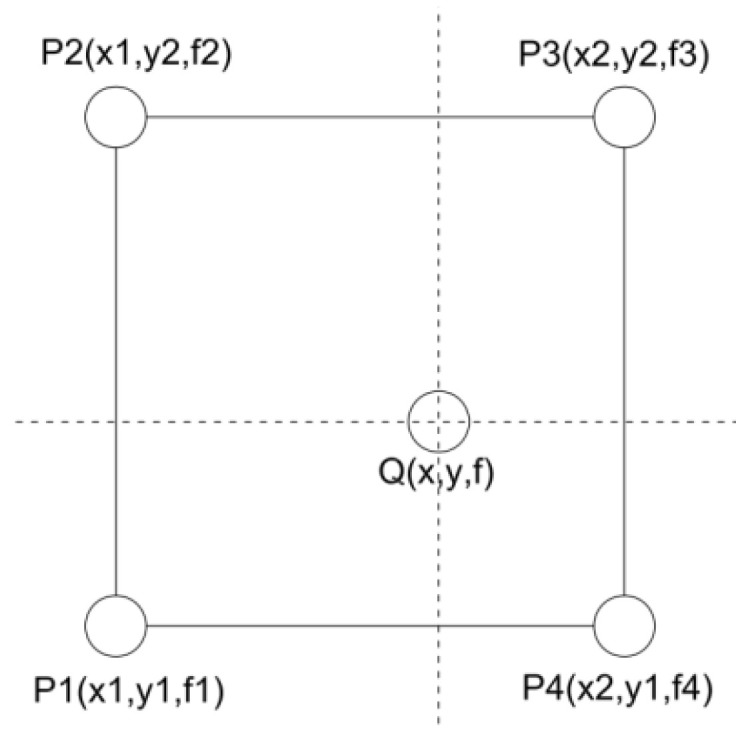
Schematic diagram of bi-linear interpolation.

**Figure 6 sensors-22-09366-f006:**
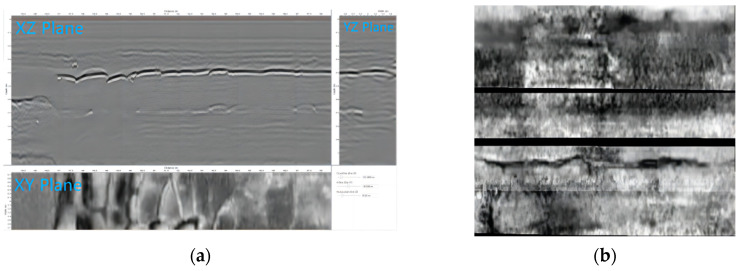
Splicing results of three slices and planes of radar images. (**a**) Schematic diagram of X–Y section, Y–X section, and X–Z section. (**b**) Mosaic image of multiple test lanes on X–Y section.

**Figure 7 sensors-22-09366-f007:**
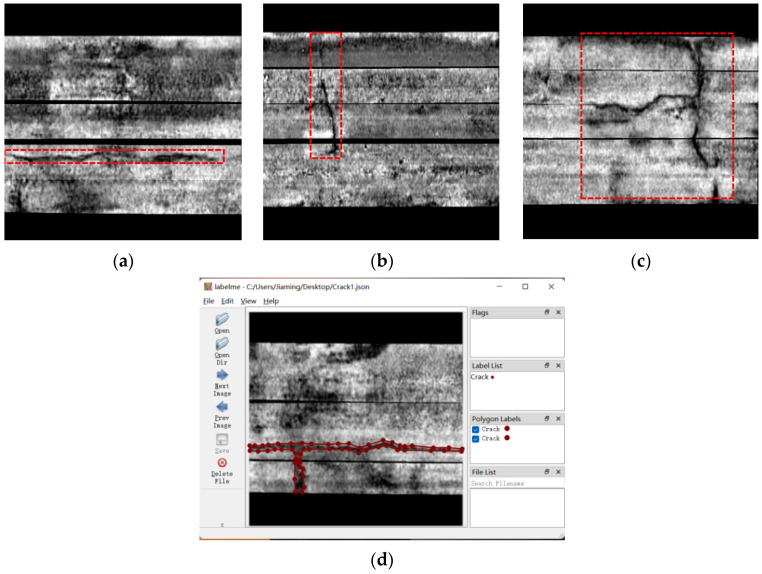
Three types of crack fractures (labeled by the red boxes) and the labeling process. (**a**) Longitudinal crack. (**b**) Transverse crack. (**c**) Cross crack. (**d**) Labelme labeling process.

**Figure 8 sensors-22-09366-f008:**
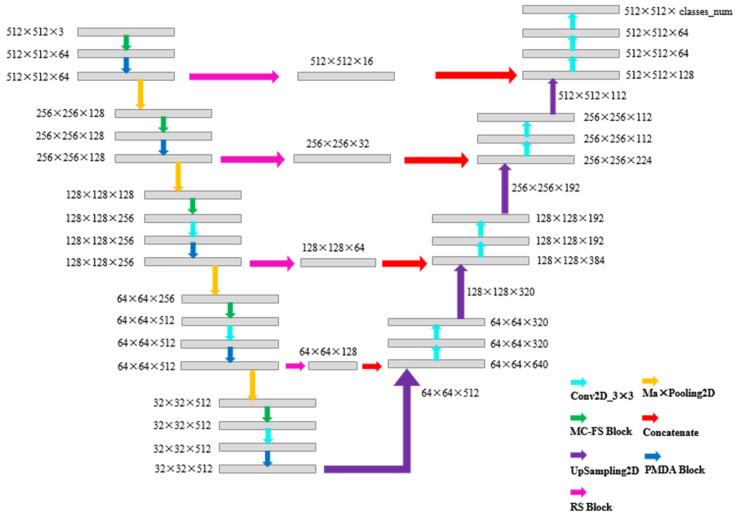
Crack UNet model structure.

**Figure 9 sensors-22-09366-f009:**
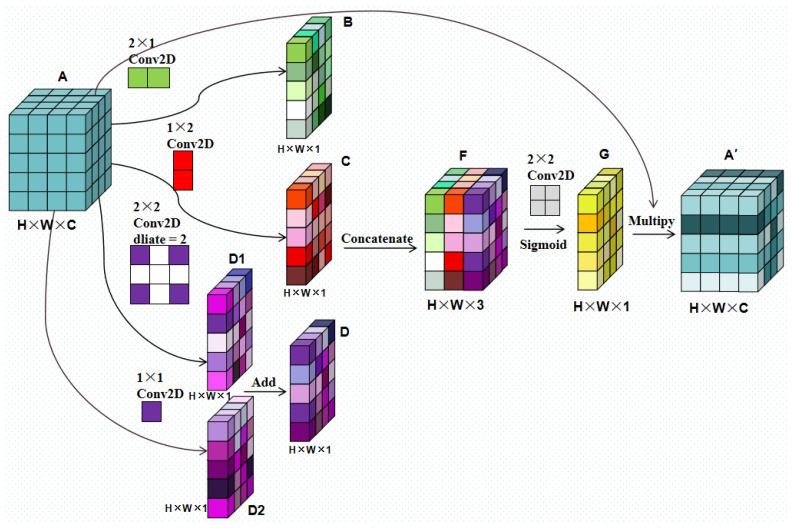
PDMA mechanism structure.

**Figure 10 sensors-22-09366-f010:**
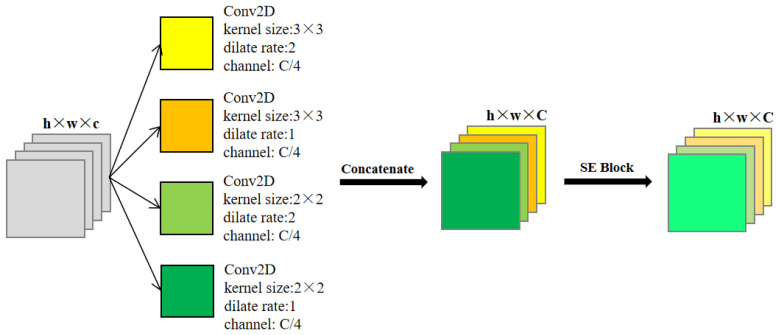
Multi-channel convolution feature selection module structure.

**Figure 11 sensors-22-09366-f011:**
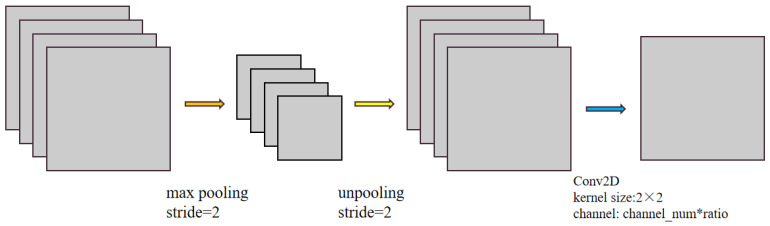
Residual compression module.

**Figure 12 sensors-22-09366-f012:**
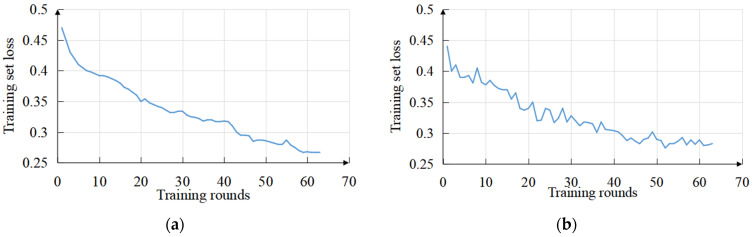
Loss function curve. (**a**) Training set loss. (**b**) Validation set loss.

**Figure 13 sensors-22-09366-f013:**
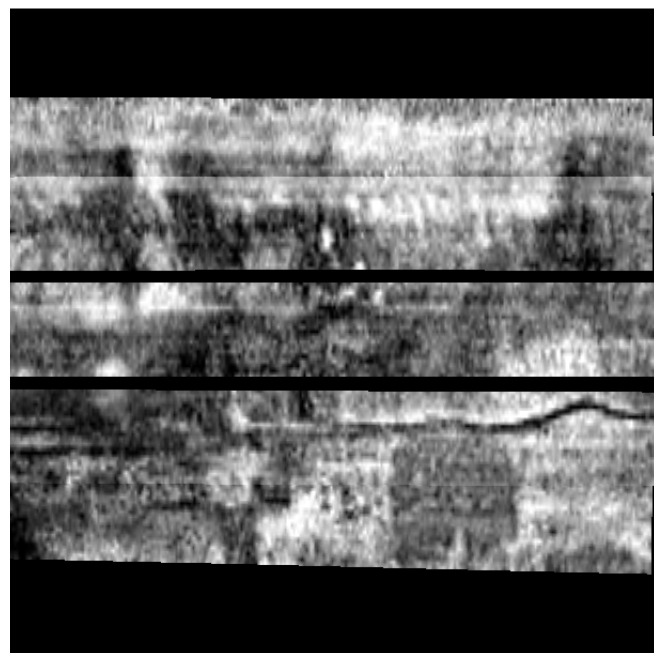
Crack images with blurry boundaries.

**Figure 14 sensors-22-09366-f014:**
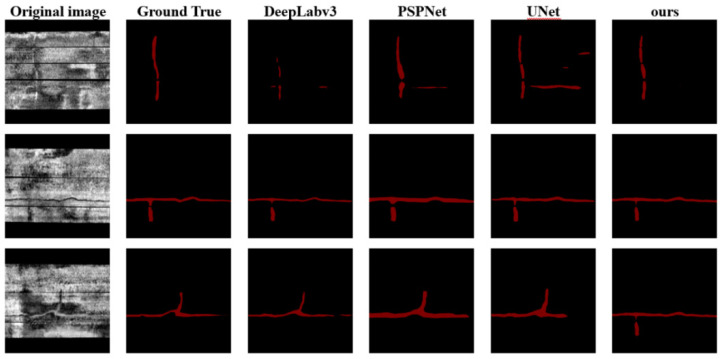
Comparison of segmentation diagrams among models in crack test set.

**Figure 15 sensors-22-09366-f015:**
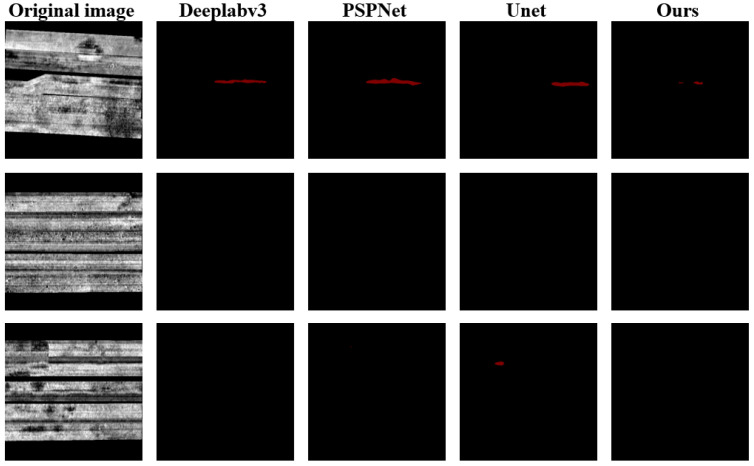
Segmentation results from non-crack data set.

**Table 1 sensors-22-09366-t001:** The explains of the processing steps of Figure 8.

Processing Steps	Explains	Figure of the Structure
Conv2D	This is a general two-dimensional convolution process, and its processing parameters are set as follows: Kernel size: 3 × 3; Stride = 1; Padding = 2.	\
MaxPooling2D	MaxPooling2D takes the maximum value of the pooled core, and its processing parameters are set as follows: Kernel size: 2 × 2; Stride = 2.	\
PMDA Block	The parallel multi-direction attention (PMDA) mechanism proposed herein is a spatial attention mechanism, whose principle is to break down a 3 × 3 convolution kernel into four parts, and then, correlate a pixel with its adjacent pixels artificially from the structure.	Figure 9
MC-FS Block	The multi-channel convolution feature selection module (MC-FS Module) can provide diverse features and give higher weight to effective features from the dimension of the channel. The MC-FS module consists of parallel convolution, extended convolution, and channel attention modules.	Figure 10
Concatenate	The concatenate treatment is to superimpose multiple multi-dimensional features into the higher dimensional feature map.	\
UpSampling2D	The purpose of upsampling is to upsample the image into the higher resolution. The research adopted a Zero-Complement up-sampling method. Its processing parameters are set as follows: Kernel size: 2 × 2; Stride = 1.	\
RS Block	The residual compression module (RS) was designed to compress the feature channel and reduce the parameter number of network model by a quarter. It consists of three steps: MaxPooling, UpSampling and Conv2D. Its processing parameters are set as follows: Kernel size: 2 × 2; Stride = 2; Padding = 2.	Figure 11

**Table 2 sensors-22-09366-t002:** Comparison of indexes from different models.

Model	Input Shape	MPA	MIoU	FLOPs	FPS
deeplabv3	512 × 512	78.34%	73.62%	83.22 M	16.76
PSPNet	473 × 473	83.42%	74.03%	98.46 M	23.93
Unet	512 × 512	82.09%	72.75%	49.77 M	23.38
Crack Unet	512 × 512	85.69%	75.02%	44.06 M	18.60

**Table 3 sensors-22-09366-t003:** The segmentation results of each model in the crack test set and the quantity distribution in each coverage interval.

Intervals	DeepLabv3	Unet	pspNet	Crack Unet
[0, 0.1)	3	1	0	1
[0.1, 0.2)	6	3	2	2
[0.2, 0.3)	15	15	6	3
[0.3, 0.4)	24	20	12	13
[0.4, 0.5)	59	26	58	16
[0.5, 0.6)	85	56	32	33
[0.6, 0.7)	110	80	53	64
[0.7, 0.8)	94	124	95	108
[0.8, 0.9)	76	116	142	130
[0.9, 1.0]	28	59	100	130
Accuracy	78.6%	87.0%	84.4%	93.0%

**Table 4 sensors-22-09366-t004:** The segmentation accuracy of each model under different false positive proportion thresholds using the non-crack test set.

Intervals	DeepLabv3	Unet	pspNet	Crack Unet
<0.1%	92.80%	53.80%	90.80%	88.80%
<0.5%	98.20%	80.40%	95.80%	94.40%
<1%	99.40%	93.40%	98.53%	97.20%

**Table 5 sensors-22-09366-t005:** Comparison of ablation experiment results.

Model	MPA	MIoU	FLOPs	FPS
Baseline	82.09%	72.75%	49.77 M	23.38
Baseline + PMDA	83.80%	73.68%	49.78 M	21.10
Baseline + MC-FS	85.45%	73.55%	51.06 M	20.69
Baseline + RS	83.79%	74.38%	42.75 M	23.36
Baseline + PMDA + MC-FS	83.95%	74.61%	51.08 M	18.65
Baseline + PMDA + RS	83.63%	74.28%	42.76 M	20.74
Baseline + MC-FS + RS	84.30%	74.68%	44.06 M	18.46
Crack Unet	85.69%	75.02%	44.06 M	18.60

**Table 6 sensors-22-09366-t006:** Test results from different attention mechanisms on crack test set.

Attention Block	MPA	MIoU	FLOPs	FPS
Crack Unet (SENet)	84.32%	74.77%	45.17 M	19.12
Crack Unet (CBAM)	84.72%	74.85%	47.55 M	16.75
Crack Unet (PMDA)	85.69%	75.02%	44.06 M	18.60
Unet (SENet)	83.18%	73.56%	50.89 M	21.80
Unet (CBAM)	83.58%	73.45%	53.27 M	18.6
Unet (PMDA)	83.80%	73.68%	49.78 M	21.02
Crack Unet (SENet)	84.32%	74.77%	45.17 M	19.12
Crack Unet (CBAM)	84.72%	74.85%	47.55 M	16.75

## Data Availability

Not applicable.

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
