# Peer review of "Crack Unet: Crack Recognition Algorithm Based on Three-Dimensional Ground Penetrating Radar Images"

_sensors, 2022, doi:10.3390/s22239366_

Round 1
Reviewer 1 Report
The authors proposed an improved Unet semantic segmentation model based on VGG16 for 3D ground-penetrating radar image crack processing. The methodology was described comprehensively and the results are meaningful. It is well done. Only a few of problems exist in this paper which the authors must pay attention to deal with.
1. The significance of this study should be highlighted in the end of the abstract.
2. Please offer the full name for the term which is mentioned at the first time. For example, the 2D, 3D, and CT etc..
3. In the Introduction, the literature review is not comprehensive. Please study the previous research on this topic with regards to the experimental and data analysis in the past 5 years. Pay attention to some related literatures from the research groups of Prof. Zhen Leng and Prof. Imad L Al-Qadi.
4. Some expressions in the manuscript are not clear. An English native speaker is suggested to carefully proofread it again. And some mistakes can be found as well. Please carefully check through the manuscript. For example, in lines 47-48, “It has significantly reduced the misjudgment rate and therefore become an effective approach to detect the cracks and damages inside pavement structure.” “Become” should be “becomes”. In line 237, parallel multi-direction attention (PDMA) should be PMDA. In the lines 336-337, “Error! Reference source not found”.
5. The equations should be numbered continuously.
Author Response
Dear Editors and Reviewers:
Thank you for your letter and for the reviewers’ comments concerning our manuscript"entitled “Crack Net: Crack Recognition Algorithm Based on Three-dimensional Ground Penetrating Radar Images"(ID:1987579). Those comments are all valuable and very helpful for revising and improving our paper, as well as the important guiding significance to our researches. We have studied comments carefully and have made correction which we hope meet with approval. Revised portion are marked up using the “Track Changes” function in the paper. The main corrections in the paper and the responds to the reviewer's comments are as flowing:
Reviewer #1:
The authors proposed an improved Unet semantic segmentation model based on VGG16 for 3D ground-penetrating radar image crack processing. The methodology was described comprehensively and the results are meaningful. It is well done. Only a few of problems exist in this paper which the authors must pay attention to deal with.
1.The significance of this study should be highlighted in the end of the abstract.
Response: We are very sorry that we have neglected to highlight the significance of this study in the end of the abstract. Relevant content has been further supplemented. As shown in the revised manuscript Page 1 - lines 23 to 27.
2.Please offer the full name for the term which is mentioned at the first time. For example, the 2D, 3D, and CT etc.
Response: Considering the Reviewer’s suggestion, Three-dimensional(3D), Computer Tomography(CT), Two-dimensional(2D) have been modified where they are mentioned at the first time.As shown in the revised manuscript Page 1 - lines 10, Page 2 - lines 41 and line 51.
3.In the Introduction, the literature review is not comprehensive. Please study the previous research on this topic with regards to the experimental and data analysis in the past 5 years. Pay attention to some related literatures from the research groups of Prof. Zhen Leng and Prof. Imad L Al-Qadi.(Prof. Zhen Leng and Prof. Imad L Al-Qadi no paper)
Response: We have re-written the literature review according to the Reviewer's suggestion. The review of studies on the application of convolutional neural networks to GPR image interpretation in the past 5 years include CNN model, 3-D convolutional neural network, MRF-UNet, A-Unet, et, al have been supplemented in the paper. I have looked up the related literatures from the research groups of Prof. Zhen Leng and Prof. Imad L Al-Qadi carefully, their studies are excellent and I cited them in the introduce.As shown in the revised manuscript Page 21 - lines 667 to 670.
4.Some expressions in the manuscript are not clear. An English native speaker is suggested to carefully proofread it again. And some mistakes can be found as well. Please carefully check through the manuscript. For example, in lines 47-48, “It has significantly reduced the misjudgment rate and therefore become an effective approach to detect the cracks and damages inside pavement structure.” “Become” should be “becomes”. In line 237, parallel multi-direction attention (PDMA) should be PMDA. In the lines 336-337, “Error! Reference source not found”.
Response: We are very sorry for our incorrect writing. "Become" has been revised to “becomes”. As shown in the revised manuscript Page 2 -lines 55. "PDMA" has been revised to “PMDA”. As shown in the revised manuscript Page 10 -lines 296. The manuscript has been carefully checked through.
5.The equations should be numbered continuously.
Response: We are very sorry for our incorrect equation numbers and they have been renumbered in the revised manuscript.
Special thanks to you for your good comments.
Reviewer 2 Report
This manuscript introduces a model for 3-D GPR crack image processing. It looks that only application of Unet in GPR images, without novel idea or results. It’s not suitable for publication at the current status. Some comments are listed as follow,
1. The parameters of 3-D GPR system should be introduced;
2. Is it necessary to use deep learning methods to detect crack damage in 3-D GPR data?
3. The traditional processing methods of 3-D GPR should be introduced and compared with deep learning methods.
4. The pre-processing of GPR data should be described in details.
5. Fig1 (c) is missing, Figure (2) is not easily understood.
6. When describing the loss function, the superposition of loss functions in two directions is understated and a little confused.
7. Page 4, Line 151, what is the equation meaning? and is it equation 1? There is another equation 1 in page 10.
8. The coverage rate is the same as the recall formula in equation 8, please check it.
Overall, this manuscript is not well prepared.
Author Response
Dear Editors and Reviewers:
Thank you for your letter and for the reviewers’ comments concerning our manuscript"entitled “Crack Net: Crack Recognition Algorithm Based on Three-dimensional Ground Penetrating Radar Images"(ID:1987579). Those comments are all valuable and very helpful for revising and improving our paper, as well as the important guiding significance to our researches. We have studied comments carefully and have made correction which we hope meet with approval. Revised portion are marked up using the “Track Changes” function in the paper. The main corrections in the paper and the responds to the reviewer's comments are as flowing:
Reviewer #2:
This manuscript introduces a model for 3-D GPR crack image processing. It looks that only application of Unet in GPR images, without novel idea or results. It’s not suitable for publication at the current status. Some comments are listed as follow,
- The parameters of 3-D GPR system should be introduced;
Response: It is really true as Reviewer suggested that the parameters of 3-D GPR system should be introduced. The parameters of 3-D GPR system has been introduced. As shown in the revised manuscript Page 4 -lines 159-163.
2.Is it necessary to use deep learning methods to detect crack damage in 3-D GPR data?
Response: After careful investigation and consideration, I suppose that it’s necessary to use deep learning methods to detect crack damage in 3-D GPR data because of excellent generalization ability of deep learning methods. Other crack detection methods like Canny edge detection, they have excellent recognition performance in the case that the crack boundary is clear because it detects the crack by detecting the local gradient change in images,but in some cases, their accuracy will be severely degraded when the crack boundary is obscure. Others, like SVM requires a large amount of training samples for image feature selection and model parameter optimization when the image signal changes greatly.The excellent generalization ability is the main reason why choose deep learning methods.
- The traditional processing methods of 3-D GPR should be introduced and compared with deep learning methods.
Response: The traditional processing methods of 3-D GPR data has been introduced in the revised manuscript Page 2 -lines 91-103. Compared with deep learning methods, the main shorting of traditional processing methods of 3-D GPR data is lacking of generalization ability.
- The pre-processing of GPR data should be described in details.
Response: It is really true as Reviewer suggested that the pre-processing of GPR data should be described in details. The introduction of the pre-processing of GPR data is supplemented in the paper. As shown in the revised manuscript Page 5 -lines 190-197.
- Fig1 (c) is missing, Figure 2 is not easily understood.
Response: We are very sorry that we have neglected to add Fig1 (c). Fig1 (c) has been added and cited in the revised manuscript Page 4 -lines 162. An explanation of Figure 2 has been supplemented in the revised manuscript Page 4 -lines 167-175.
- When describing the loss function, the superposition of loss functions in two directions is understated and a little confused.
Response: Dice Loss was initially adopted to solve the problem of sample imbalance in medical image segmentation. Focal Loss is also employed to solve the problems of unbalanced training samples and with different difficulties in sample, which is a variant of cross entropy.
- Page 4, Line 151, what is the equation meaning? and is it equation 1? There is another equation 1 in page 10.
Response: Why they were added together because in the course of the experiment, We found that the recognition accuracy of the trained model was higher when they were added than when they were arbitrarily used as loss functions alone.The two functions should complement each other in some way, but it is difficult to explain how they work. It is difficult for deep learning to explain its operating principle clearly and rigorously.
- The coverage rate is the same as the recall formula in equation 8, please check it.
Response: We are very sorry for our incorrect equation numbers and they have been renumbered in the revised manuscript.The equation 6, 7 and 9 are not same.
Special thanks to you for your good comments.
With best regards,
Sincerely Yours,
Jiaming Tang
Reviewer 3 Report
This paper proposed an improved Unet semantic segmentation model named Crack Unet based on VGG16 for 3D ground-penetrating radar image crack processing. The experimental results demonstrated that the Crack Unet model has the better performance in the radar image crack segmentation task than the current mainstream algorithms.
There are some comments as follows.
1) In the Figure 1, there is missing content in the box of subfigure b, and the subfigure C is missing.
2) In the experiments, the detailed parameters of networks should be provided.
3) According to the title, the 3D GPR images are focused to implement crack recognition. However, in the main text, only the 2D slice images in X-Y plane are processed to recognize the cracks. Please make the description clearer.
4) As we know, the cracks not only exist in X-Y plane, but also emerge in the X-Z plane and Y-Z plane. However, this method only deals with the problem of crack recognition in X-Y plane. Could you please provide the detailed explanation.
5) Before the crack recognition in X-Y plane, how to determine the location of crack in the X-axis, ensuring the accurate extraction of slice image in X-Y plane? How to implement the extraction of slice images in X-Y plane in the case of multiple cracks?
6) In the Equation (2), how to set a optimal value for pt?
7) In the lines 336 and 337, the references are missing.
Author Response
Dear Editors and Reviewers:
Thank you for your letter and for the reviewers’ comments concerning our manuscript"entitled “Crack Net: Crack Recognition Algorithm Based on Three-dimensional Ground Penetrating Radar Images"(ID:1987579). Those comments are all valuable and very helpful for revising and improving our paper, as well as the important guiding significance to our researches. We have studied comments carefully and have made correction which we hope meet with approval. Revised portion are marked up using the “Track Changes” function in the paper. The main corrections in the paper and the responds to the reviewer's comments are as flowing:
Reviewer #3:
This paper proposed an improved Unet semantic segmentation model named Crack Unet based on VGG16 for 3D ground-penetrating radar image crack processing. The experimental results demonstrated that the Crack Unet model has the better performance in the radar image crack segmentation task than the current mainstream algorithms.
There are some comments as follows.
1.In the Figure 1, there is missing content in the box of subfigure b, and the subfigure C is missing.
Response: We are very sorry that we have neglected to add Fig1 (c). Fig1 (c) has been added and cited in the revised manuscript Page 4 -lines 162. An explanation of Figure 2 has been supplemented in the revised manuscript Page 4 -lines 167-175.
2.In the experiments, the detailed parameters of networks should be provided.
Response: It is really true as Reviewer suggested that we should provide the detailed parameters of networks. The detailed parameters of networks have been supplemented in the revised manuscript Page 13 -lines 386-400. Because step length and receptive field are different in different modules, they have to been found in the figure 8-11 of Part 4, Crack Unet Algorithm.
3.According to the title, the 3D GPR images are focused to implement crack recognition. However, in the main text, only the 2D slice images in X-Y plane are processed to recognize the cracks. Please make the description clearer.
Response: In the title, We emphasize 3D ground penetrating radar because the image of cracks is relatively easy to find in X-Y plane but not in the Y-Z plane and X-Z plane. The 2D GPR is hard to get X-Y plane because what the 2D GPR detects is Y-Z plane, they have to splice many Y-Z plane to C-scan and separate the X-Y plane from C-scan. But the C-scan collected on this way is hard to be uniform in general because the distance between different Y-Z planes is hard to be uniform.Moreover, collecting the C-scan using 2D GPR on this way takes a lot of time and effort. That is why we emphasize 3D ground penetrating radar in the title meanwhile only the 2D slice images in X-Y plane are processed.
4.As we know, the cracks not only exist in X-Y plane, but also emerge in the X-Z plane and Y-Z plane. However, this method only deals with the problem of crack recognition in X-Y plane. Could you please provide the detailed explanation.
Response: As the crack signal performed in the three perpendicular slices was shown as it was in Figure 6 (a). It can be seen that the crack signal is not obvious on the YZ plane and XZ plane but is relatively obvious on the XY plane. That is related to the longer length and the generally narrow width and low height of crack as shown in the revised manuscript Page 7 -lines 209-213. In my opinion, the shape of the crack is like a flat snake, and it is not obvious to cut horizontally or vertically. It is only when cut flat that the cross section is the largest and the shape of the crack is the most obvious.
5.Before the crack recognition in X-Y plane, how to determine the location of crack in the X-axis, ensuring the accurate extraction of slice image in X-Y plane? How to implement the extraction of slice images in X-Y plane in the case of multiple cracks?
Response: We are very sorry that we have neglected to explain the method of location determination. The location of crack in the X-axis is mainly according to is position in pixel and every pixel means 0.009m as shown in the revised manuscript Page 6 -lines 198-202. The Crack Unet categorizes each pixel in an image so as to segment the crack accurately so that it can solve the problem of the extraction of slice images in X-Y plane in the case of multiple cracks.
6.In the Equation (2), how to set a optimal value for pt?
Response: As shown in the revised manuscript Page 6 -lines 197-200, in order to homogenize the data matrix, it’s necessary to interpolate the horizontal (y) and traveling direction (x) so that the spacing in those two directions is close to the depth direction and that the spacing among the points in the three dimensions of x, y and z is 0.009 m. So, the distance of the pixels is determined by the smallest distance in three dimensions.And the radar data value of point inserted is calculated according to four corner points by equation 1.
7.In the lines 336 and 337, the references are missing.
Response: We are very sorry for our incorrect writing. Content has been modified as shown in the revised manuscript Page 13 -lines 386-400.
Special thanks to you for your good comments.
With best regards,
Sincerely Yours,
Jiaming Tang
Round 2
Reviewer 2 Report
This manuscript should be reorginized and then resubmit.
1) the contribution is not clear;
2) the figures are not well drawn (fig 4, fig 7, fig 8, fig 12);
3) the theory and method are not written in the right way, including equations and flow charts.
Author Response
Dear Editors and Reviewers:
Thank you for your letter and for the reviewers’ comments concerning our manuscript"entitled “Crack Net: Crack Recognition Algorithm Based on Three-dimensional Ground Penetrating Radar Images"(ID:1987579). Those comments are all valuable and very helpful for revising and improving our paper, as well as the important guiding significance to our researches. We have studied comments carefully and have made correction which we hope meet with approval. Revised portion are marked up using the “Track Changes” function in the paper. The main corrections in the paper and the responds to the reviewer's comments are as flowing:
Reviewer #2:
This manuscript should be reorginized and then resubmit.
- the contribution is not clear;
Response: We are very sorry that we have not explain the contribution of the article clearly. The main problem solved in this paper is the automatic interpretation of GPR images.Ground penetrating radar (GPR) has been widely used in crack damage detection of pavement structures, but the interpretation of GPR images is mainly manual interpretation. But many problems exist in manual radar image interpretation. For example, the interpretation process requires highly professional expertise, and there is a lack of data interpreters. The interpretation process is subjective; a same radar image often produces different results when it is interpreted by different people. The manual interpretation process is featured by its long period, huge workload, and low efficiency. Those disadvantages have hindered the application and promotion of ground-penetrating radar technology to a certain degree.
Therefore, an improved Crack Unet model based on Unet semantic segmentation model was proposed herein to tackle the problems in 3D ground-penetrating radar image crack recognition.Through the steps of data set construction, model optimization design and test verification, the results show that the performance of the model is improved after optimization, indicating that the optimization design is reasonable. The performance of the Crack Unet model is better than the mainstream models when they interprete the same data set. Moreover, the accuracy of Crack Unet model reaches 93.0%, which basically meets the engineering requirements of crack segmentation of radar images. And in the end, we conclude the contribution that the Crack Unet model has excellent engineering application value in the task of crack segmentation of radar images.
To illustrate the contribution of the article, I added a paragraph to the conclusion as shown in the revised manuscript Page 4 -lines 642-648.
- the figures are not well drawn (fig 4, fig 7, fig 8, fig 12);
Response: We are very sorry that the figures are drawn not well. The Reviewer has not indicated the specific problems in the figures. I did a thorough search for the problems with the figures. I found the problems of figures and took the improvements as follow:
fig 4: I think the problem of Figure 5 is that the text in Figure 5 is not clear. Therefore, I re-captured the Figure 4, the parameter settings of preprocessing, as shown in the revised manuscript Page 6 -Figure 4.
fig 7: I think the problem of Figure 7 is that the text in Figure 7 (d) is not clear and the title of Figure 7 is wrong. Therefore, I re-captured the Figure 7 (d), the operating interface of labelme, as shown in the revised manuscript Page 7 -Figure 7.
fig 8: I think the main problem of Figure 8 is that the processing steps are not easy for readers without a knowledge background in artificial intelligence to comprehend because some processing steps are not well explained. Therefore, I made an extra chart to explain each step, as shown in the revised manuscript Page 9 -Table 1.
fig 12: I think the main problem of Figure 12 is that the line chart lacks coordinate axis names. Therefore, I redrew the picture, as shown in the revised manuscript Page 14 -Figure 12.
3) the theory and method are not written in the right way, including equations and flow charts.
Response: We are very sorry for our incorrect writing. We have carefully checked all the formulas and the interpretation of the symbols in the formulas, and found that the expression is not clear and the format was not written in the right way. They been corrected. Is what the flow chart you mean figures 8 to 11? These four figures are drawn by referring to the figure styles of some articles on artificial intelligence model algorithm. They are as follows:
1.Weng, W.; Zhu, X. UNet: Convolutional Networks for Biomedical Image Segmentation, IEEE Access, 2021, PP(99), 1-1, DOI:10.1109/ACCESS.2021.3053408.
2.Chen, L.C.; Papandreou, G.; Schroff, F.; et al. Rethinking Atrous Convolution for Semantic Image Segmentation, 2017, DOI:10.48550/arXiv.1706.05587.
3.Badrinarayanan, V.; Kendall, A.; Cipolla, R. SegNet: A Deep Convolutional Encoder-Decoder Architecture for Image Segmentation, IEEE Transactions on Pattern Analysis & Machine Intelligence, 2017, 1-1, DOI:10.1109/TPAMI.2016.2644615.
Maybe the flow charts are not very good, but I can't think of a better way to write them right now.
Special thanks to you for your good comments.
We tried our best to improve the manuscript and made some changes in the manuscript. These changes will not influence the content and framework of the paper. And here we did not list the changes but marked in red in revised paper.
We appreciate for Editors/Reviewers' warm work earnestly, and hope that the correction will meet with approval.
Once again, thank you very much for your comments and suggestions.
With best regards,
Sincerely Yours,
Jiaming Tang
Reviewer 3 Report
The authors have addressed all my concerns, so I reccomend the paper for publication.
Author Response
Dear Reviewer,
Thank you very much for your report.
With best regards,
Sincerely Yours,
Jiaming Tang